# The Integrated Effect of Environmental Conditions and Human Presence on the Behaviour of a Pair of Zoo-Housed Asian Small-Clawed Otters

**DOI:** 10.3390/ani13132228

**Published:** 2023-07-06

**Authors:** Francesca Bandoli, Jenny Mace, Andrew Knight

**Affiliations:** 1Giardino Zoologico di Pistoia, Via Pieve a Celle 160a, 51100 Pistoia, Italy; 2Centre for Animal Welfare, Faculty of Health and Wellbeing, University of Winchester, Sparkford Road, Winchester SO22 4NR, UK; 3School of Environment and Science, Nathan Campus, Griffith University, 170 Kessels Rd, Nathan, Brisbane, QLD 4111, Australia

**Keywords:** *Aonyx cinereus*, animal welfare, human–animal interaction, animal behaviour, otter welfare, zoo welfare

## Abstract

**Simple Summary:**

Various studies have assessed animal welfare by analysing how behaviour is affected by environmental and human-related factors. Nevertheless, the combined effects of these factors are poorly researched. This study assessed for the first time the integrated impact of environmental conditions and visitor and caregiver presence on the behaviour of a pair of zoo-housed Asian small-clawed otters. We collected data across 14 sampling days from July to September 2020. We video-recorded the otters’ behaviours using continuous focal animal sampling (42 h of observation/subject). We found that the otters only performed species-specific behaviours and mainly experienced positive welfare states. However, they dedicated less time to locomotion, food-related and affiliative behaviours, and engaged more in vigilance compared to previous studies carried out in captive settings. Interactions between the otters and visitors/caregivers were limited and mostly associated with neutral or positive behavioural responses. Visitor presence and background noise did not affect behaviour. Time of day and animal identity influenced locomotion, vigilance, time spent out of sight, juggling, and visitor–otter interactions. Our results provided meaningful information to improve husbandry practices and highlighted the need to conduct multivariate analysis to better assess the welfare of animals under human care.

**Abstract:**

Zoos and aquaria have the ethical responsibility to provide animals under their care with conditions that promote good welfare. This study evaluated the combined influence of human presence and environmental factors on the behaviour of zoo-housed Asian small-clawed otters (*Aonyx cinereus*). Data collection was conducted on a pair hosted at Pistoia Zoo (Italy). Data were collected from July to September 2020 (over 14 days). We video-recorded the otters’ behaviours, using the continuous focal animal sampling, obtaining 42 h of observation per subject. The otters displayed a wide array of species-specific behaviours. Compared to previous captive studies, the subjects engaged less in locomotion, food-related and affiliative behaviours, and more in vigilance. Human–animal interactions were limited, and mostly elicited neutral or positive responses, except for begging behaviours performed towards caregivers. Time of day and animal identity were the main significant predictors for behaviours. No effects of visitor presence or background noise were detected. Nonetheless, increasing enrichment use could stimulate food-related behaviours, and reduce vigilance and begging. This study confirms the importance of applying an integrated approach to analyse the complexity of otters’ experiences, and provides insights to guide husbandry improvements.

## 1. Introduction

Zoos and aquaria worldwide collectively hold approximately 600,000 animals [1] and, according to the available data, they are visited by over 700 million people each year [2]. Their aims include the preservation of wildlife through research, educational and conservation activities, while providing good welfare conditions to the animals in their care [3]. Nonetheless, due to the variety of species and individual animals with different histories, needs and preferences, ensuring high welfare standards is complex in terms of resources and the diverse knowledge required [4,5].

Advances in husbandry practices and zoo animal welfare science are key to guaranteeing that zoological institutions work as scientific and ethical conservation centres [6]. However, as emphasised by different scholars [7,8], more studies concerning welfare assessment are required in many areas to examine and enhance captive animal management. Furthermore, since welfare is linked to individual survival and reproduction success [9], these studies also help to maximise the impact of conservation projects based on ex situ and in situ breeding and wild animal reintroduction programs [5,10].

### 1.1. Zoo Animal Welfare

According to Mellor and colleagues [11], an animal’s welfare state can be described as the balance between its negative and positive subjective experiences. Animals are deemed to reach a positive welfare state when they mostly experience positive affects (e.g., a sense of security, sometimes indicated by playfulness), and when their physical and behavioural needs are met, and they can exert choice and control over resources and the environment.

Lack of space and enclosure complexity, repeated negative interactions with caregivers and visitors, and limited opportunities to choose among different resources, to decide where and when to perform behaviours, and with whom to interact, are among the main problems that can compromise the welfare of animals under human care [12,13,14]. Results may include a reduction in species-specific behaviours associated with positive emotional states (e.g., affiliative behaviour), and the increase in negative ones, for example, aggressive behavioural patterns [13]. Moreover, they can lead to the onset of abnormal repetitive behaviours (ARBs), i.e., repetitive, unvarying and seemingly functionless behavioural patterns [15], which are usually interpreted as an indicator of poor welfare either currently or in the past [16]. However, due to the lack of an in-depth knowledge of the perceptual, sensory and emotional capabilities of many wild species, understanding what captive animals might be experiencing is not straightforward [5].

Many studies have tried to evaluate animal welfare by verifying whether and how environmental and human stimuli (e.g., ambient temperature and visitor presence, respectively) affect animals’ behaviour [8]. Nevertheless, only a few researchers have examined the combined impacts of these stimuli in the attempt to obtain a clearer picture of animals’ experiences (see for example: [17,18,19,20,21]). Interestingly, some of these studies detected an overestimation of the visitor effect (i.e., the positive or negative influence exerted by visitor presence) when other parameters were included in the analysis. For example, Goodenough et al. [17] found that weather conditions and time of day influenced the behaviour of a troop of ring-tailed lemurs (*Lemur catta*) hosted at West Midlands Safari Park (Bewdley, UK) more than visitor presence. Rose and colleagues [18] discovered, instead, that ambient temperature, irrespective of visitor numbers, accounted for the behavioural changes recorded in a pair of black-casqued hornbills (*Ceratogymna atrata*) living at Blackpool Zoo (Blackpool, UK). Riley and co-authors [19] analysed the behaviour of 22 Nile crocodiles (*Cocodrylus niloticus*) housed at Disney’s Animal Kingdom (Bay Lake, FL, USA), collecting data with and without visitors present. Ambient temperature, time of day and month were found to significantly influence behaviour, whereas visitor presence had a neutral effect. These results suggest that a more holistic methodological approach, based on the integration of a wide array of factors, is needed to properly tackle the complexity of the zoo environment and provide valuable information to improve animal welfare.

### 1.2. Asian Small-Clawed Otters

Asian small-clawed otters (hereafter referred to as ASCOs) are found in the wild in India, Nepal, Bhutan, Southern China and Southeast Asia [22,23], and are small semi-aquatic mammals with an average weight of 3 kg [24]. They inhabit natural and human-altered environments, ranging from meandering rivers to rice fields [22]. They are mainly active during the day [23], and their diet mostly consists of crabs and shellfish [25]. They live in family groups (comprising, on average, five individuals) based on a dominant pair that can breed year-round [24]. They seem to be monogamous [26], and have been reported to live from two to six years of age in the wild and up to 20 years in captivity [24]. Due to their elusive nature, little is known about their behaviour in the wild [23].

Since 2008, the ASCO has been classified as ‘vulnerable’ in the Red List of Threatened Species issued by the International Union for Conservation of Nature [22]. Its survival is threatened by habitat loss, environmental pollution, conflict with local communities, and poaching [23]. The species is protected in part of its range [23], and in 2019 it was listed in Appendix I of the Convention on International Trade in Endangered Species of Wild Fauna and Flora (CITES) and its commercial trade was banned [27].

ASCOs are commonly bred in captivity [24], and each individual is recorded in an international studbook coordinated by the World Association of Zoos and Aquariums [28]. Moreover, the American (AZA) and the European Association of Zoos and Aquaria (EAZA), respectively, manage a regional studbook and an ex situ breeding program to maintain viable insurance populations within zoological facilities [24,29]. While *A. cinereus* is the most common otter species housed under human care [30,31], to date, a limited number of studies have focused on otter welfare assessment. As impetus for further study, previous research on ASCOs reported conflicting results concerning time budgets, and often detected the occurrence of ARBs [30,32,33,34,35,36,37,38,39].

In the available literature, the time budget was found to differ in relation to group composition and housing styles. Studying family groups in outdoor naturalistic exhibits at different UK institutions, Owen [35]), Gothard [36], and Cuculescu-Santana et al. [30] discovered that approximately 30% of the time was spent out of sight. They also found that, when visible, the animals mainly engaged in locomotion, social grooming and vigilance. Research on pairs or groups of siblings kept in indoor enclosures found that they seem to be less active and mostly engage in solitary behaviours [33,39], with potential negative effects on their physical and social skills [40].

The potential influence of environmental conditions on otters’ time budgets has been evaluated in only two studies. One study involved two adult male siblings and the other one a family group of seven otters [30,37]. Both pieces of research were carried out in UK facilities and were aimed at assessing seasonal changes in the otters’ behaviour, focusing on air and water temperatures as key environmental parameters. Overall, the results showed that the subjects spent more time swimming and less time engaged in resting, vigilance and aggressive behaviours during summer. These findings suggest that providing ASCOs with at least a partially heated pool during winter could help to stimulate active behaviours across all seasons, while reducing behaviours with potentially negative welfare outcomes.

Although some patterns related to the influence of group composition, housing styles, and environmental and human-related stimuli on otter welfare can be found in the available studies, a clear and quantitative picture cannot be yet identified, due to the limited number of parameters examined, and the use of a research methodology that does not take into account the potential complex relations between the stimuli otters are exposed to.

In this study, we aimed to assess the integrated effects of environmental conditions (i.e., ambient/water temperature, relative humidity, and background noise level) and caregiver/visitor presence, on the behaviour of a pair of captive ASCOs through the application of multivariate research methods. In particular, we investigated the subjects’ behavioural time budget, the occurrence and potential causes of ARBs, and evaluated whether and how the otters’ behaviours were affected by selected environmental and human-related parameters.

According to the literature [30,33,35,36,39], we expected the subjects would allocate less time to foraging activities compared to their wild conspecifics [41]). As found in previous research carried out on ASCOs in captivity [30,33,35,36], we also expected that the otters would spend most of their time out of sight, and, when visible, that they would mainly engage in locomotion, vigilance and social behaviours. We predicted the presence of ARBs, specifically begging behaviours performed towards caregivers and visitors. Finally, we expected to find that the time of day, temperature and humidity impact the otters’ behavioural responses, with visitor presence only having a limited effect.

## 2. Materials and Methods

### 2.1. Study Subjects

This study focused on a pair of ASCOs housed at the Giardino Zoologico di Pistoia (GZP), Pistoia, Italy. The pair consisted of two unrelated individuals: a male born on 8th December 2017 at Parco Faunistico Cappeller, Vicenza, Italy, and a female born on 6th July 2016 at Zoom Torino Spa, Turin, Italy. They were reared by their parents and housed within their family groups until the transfer to GZP in December 2018. At the time of the study, they were three and four years old, respectively. Otters become sexually mature at approximately two to three years of age, and normally live up to 20 years in captivity [24]. The study subjects were, therefore, young adults. No record of illnesses was reported by the GZP staff. In line with the EAZA recommendations [24], otters can be bred once but, at the time of the study, no successful breeding events were recorded.

Routine husbandry procedures were mainly based on a hands-off approach, which requires minimising contact between caregivers and animals [42]. The otters were fed crustaceans, molluscs and fish four times per day (8:30 am, 11:00 am, 3:30 pm, and 6:00 pm). Food was usually scattered from outside the enclosure, except for the feeding at 11:00 am, when the subjects were locked and fed inside the dens (for details on the otter’s enclosure see Section 2.2, ‘Study Site’) to enable caregivers to perform daily cleaning.

### 2.2. Study Site

The host institution was licensed under Italian legislation, and was a member of the Italian and European Associations of Zoos and Aquaria. The otters were kept in an open-topped naturalistic exhibit consisting of an outdoor section of 200 m^2^, equipped with an artificial freshwater pool of 32 m^3^ with quartz sand filter pumps (Figure 1). The otters also had at their disposal an indoor area of 2 m^2^ with a wooden nest box and heating lamps, and two additional dens (60 × 60 × 50 cm^3^) with underfloor heating, connected via a PVC tunnel with the outdoor section. The perimeter was fenced with three-layer glass panels (at a height of 150 cm), concrete walls, and 50 × 50 mm welded wire mesh, equipped with an electrified perimeter wire and a 45° overhang. The staff accessed the dens and the outdoor area through a service zone of 15 m^2^. Both the dens and the indoor area were connected to the outdoor zone through sliding doors that could be operated by caregivers. The outdoor section was characterised by natural substrate and vegetation, including trunks, rocks, and a shallow stream. Seven viewing points, including two underwater viewing points, were located along three of the enclosure’s four sides. On the southern and western sides of the area a tunnel, suspension bridge, and slide were available for children. On the northern side, the area bordered the exhibit of a pair of European brown bears (*Ursus arctos*), and could not be reached by visitors or personnel. Additional fences and lush vegetation prevented the otters and bears from seeing each other. The enclosure was designed according to the available guidelines [24], and its size was 3.5 times bigger than the recommended minimum size (60 m^2^) for a pair of otters. The subjects had access to all sections of the enclosure, with the exception of the second daily feeding when they were enclosed within the dens, as described in Section 2.1 (‘Study Subjects’).

### 2.3. Data Collection

#### 2.3.1. Preliminary Study and Experimental Design

We carried out a pilot study on six non-consecutive days from 21st June to 11th July 2020, to learn how to identify the subjects and conduct preliminary observations across different time slots. We used the ‘ad libitum’ sampling method, with data samples taken at points convenient to the researcher [43]. We used data from the preliminary study to adapt pre-existing ethograms [32,37] and to develop a list of behavioural patterns specific to this study. We divided behaviours into ‘states’, i.e., behavioural patterns of relatively long duration, and ‘events’, namely behaviours which can be approximated as points in time [43]. We grouped the behaviours into categories. A total of 19 categories and 45 behaviours (40 ‘states’ and 5 ‘events’) were included in the ethogram (Table 1). It is worth noting that we divided the out of sight category into out of sight_dens_, out of sight_indoor_, and out of sight_outdoor_ to obtain information on the locations where the animals were not visible. Moreover, regarding the feeding category, we distinguished between eating/foraging and eating_dens_, to calculate the amount of time the otters were locked inside the dens in order to be fed during the second daily feeding. Similarly, we divided the abnormal repetitive behaviours category into begging and begging_outdoor_, to calculate the time the subjects were locked in the outdoor section without access to the dens when waiting to be fed by caregivers during the second daily feeding (see Section 2.1).

During the subsequent, main study, we collected data during 14 sampling days selected within a two-month period (from 19th of July to 6th of September 2020). We videoed the otters to ensure comprehensive monitoring of their behavioural patterns [44]. We conducted six one-hour observations daily (S1: 9:15–10:15 am; S2: 10:45–11:45 am; S3: 12:15–1:15 pm; S4: 2:45–3:45 pm; S5: 4:15–5:15 pm; S6: 5:45–6:45 pm) to record data evenly across the public opening hours (9:00 am–7:00 pm), and to include feeding sessions and enclosure cleaning in the data collection. We used body size, tail shape and fur colour to identify the subjects, as these parameters were sufficiently different between the two ASCOs.

**Table 1 animals-13-02228-t001:** Otter ethogram, adapted from Hawke et al. [32], Cuculescu-Santana et al. [37] and Burghardt [45].

Behavioural Category	Behaviour	Description
**Abnormal repetitive behaviours**	Begging	Standing on hind limbs with forepaws held in front of the body and repeatedly moved up and down.
	Begging_outdoor_	Begging when locked in the outdoor area.
	Head flipping	Flicking and repeated upward extension of the head and neck.
	Pacing	Moving repetitively along the same route.
	Tail biting/suckling	Repeated oral manipulation of the tail.
**Affiliative**	Allogrooming	Using paws or mouth to clean the fur of conspecifics.
	Body contact	Touching, rubbing on conspecifics’ body.
	Mutual grooming	Using paws or mouth to reciprocally clean the fur.
	Sharing food	Giving food to conspecifics.
**Agonistic**	Biting *	Using teeth to wound conspecifics.
	Chasing	Trotting or running to pursue conspecifics.
	Fighting	Rough fighting, with biting, pushing, hair plucking.
	Fleeing	Moving away from conspecifics when attacked.
	Hair plucking	Pulling the hair of conspecifics with the forepaws.
	Pushing *	Using the forepaws or body parts to displace conspecifics.
	Threatening *	Opening the mouth with teeth exposed.
**Exploratory**	Digging	Using paws to move the substrate.
	Object interaction	Manipulating, carrying inanimate items.
	Sniffing	Exploring a stimulus by inhaling air through the nose.
**Feeding**	Eating/Foraging	Moving with the head down and the nose close to the substrate. Using the mouth or forepaws to capture insects. Transporting food. Biting, chewing, handling, ingesting food.
	Eating_dens_	Biting, chewing, handling, ingesting food, while locked inside the dens.
**Human–animal interaction**	Approaching	Moving towards, following, sniffing, touching a person.
	Biting *	Using teeth to wound a person.
	Observing	Focusing the eyes on a person.
	Retreating from	Moving away from a person.
**Juggling**	Juggling	Fast, erratic movements that pass an object between the forepaws and sometimes the mouth.
**Land locomotion**	Land locomotion	Walking, running, trotting on land or flat surfaces. Climbing on higher structures.
**Maintenance**	Defecation/Urination	Eliminating urine and/or faeces.
	Drinking	Ingesting water.
	Yawning *	Opening the mouth wide to take in air.
**Mating**	Mounting	Engaging in copulatory activities.
**Nest building**	Interaction with nesting material	Manipulating, carrying nesting materials.
**Other**	Other	Performing a behaviour not listed in the other categories.
**Out of sight**	Out of sight_dens_ Out of sight_indoor_ Out of sight_outdoor_	Hidden in the dens. Hidden in the indoor area. Hidden in the outdoor area.
**Play**	Locomotor play	Intense motor activities performed in a persistent and frenetic way.
	Object play	Manipulation of inanimate items to create unpredictable situations.
	Social play	Chasing and tumbling together. Play fighting.
**Resting**	Resting	Lying down with head down.
**Scent marking**	Body rubbing	Rubbing a body part against a substrate or structure.
	Sprainting	Spreading faeces with the tail.
**Self-directed**	Self-grooming	Using the paws or mouth to clean own fur.
	Self-scratching	Self-touch with movements of the claws.
**Swimming**	Swimming	Locomotion in deep or shallow water.
**Vigilance**	Vigilance	Looking around with head up.

Note: * Asterisks represent behaviours classified as ‘events’ [43].

#### 2.3.2. Behavioural Data Recording

The otters’ behaviour was recorded by one observer using a camcorder (Panasonic HC-V180EG-K) operated from outside the enclosure, and a second camcorder (Andoer 4k Ultra HD) set up in the service area to record the otters inside one of the dens (Figure 1, DEN2). The first camcorder allowed recording of approximately 70% of the outside section of the enclosure, and the observer had to move around the enclosure’s perimeters to follow the otters’ movements. We collected data using the ‘continuous focal animal sampling’ method, which involves the continuous observation of a selected, or ‘focal’, individual/group for a specified amount of time (Martin and Bateson, 2007 [43]). Each observation session was based on a 30 min recording of each otter. We conducted a total of 84 focal sessions for each subject, resulting in 40 h and 18 min of observation for the male and 39 h and 25 min for the female. The total observation time of the female subject was slightly shorter because the observer had to interrupt one of the sessions due to technical issues. The order of the subjects was alternated to monitor them in both the first and second 30 min portion of each 1 h observation session.

#### 2.3.3. Environmental and Human-Related Parameters

Based on other similar studies [17,18,19,30,39], we selected the following parameters to be included in the analyses: weather conditions, ambient and water temperature, wind speed, relative humidity, background noise level, number of visitors, and caregiver presence. Weather conditions, wind speed, humidity, and ambient and water temperature were registered at the beginning of each observation session. Weather conditions were coded following the definitions provided by the Tuscan weather service [46]. We measured on-site wind speed with a Digital Anemometer Proster TL667SAN (Proster Trading Limited, Hong Kong, China), relative humidity and ambient temperature with a Thermohygrometer JZK TA298 (JZK, Shenzen, China), and water temperature with an Aquarium Digital Thermometer Zacro ZDT1D-AU-IT-1 (Zacro, London, UK). We registered caregiver presence in and around the enclosure, while recording visitor numbers and background noise every five minutes (seven times per focal observation). We counted visitors standing within 3 m from the fence based on a ranked score from 0 to 5: 0 = 0 people; 1 = 1–10 people; 2 = 11–20 people; 3 = 21–30 people; 4 = 31–40 people; and 5 = more than 40 people. Concurrently, we measured background noise with a Digital Sound Noise Level Meter Benetech GM1352 (Benetech, Palo Alto, CA, USA). All the measures were taken outside of the enclosure to avoid interfering with or disrupting the otters’ behaviours.

We used ambient temperature and relative humidity values to calculate the temperature–humidity index, or THI [47] which is commonly used to predict the effects of environmental warmth in farmed animals [48] (and has been employed also for zoo animals [49,50]).

We calculated the THI as follows:THI = 0:8 × T + [(RH × (T − 14:4)] + 46:4
where T is the ambient temperature in °C and RH is the relative humidity in decimal form.

### 2.4. Data Analysis

We analysed video recordings using the Behavioral Observation Research Interactive Software (BORIS), version 7.9.8 [44]. Video analysis was performed by one observer in a distraction-free environment from January to March 2021. We analysed a total of 79 h and 43 min of video recordings (as reported in Section 2.3.2 ‘Behavioural data recording’) conducting, on average, 8 h of video analysis per week. We tested intra-rater reliability using Cohen’s kappa (κ) [51], and the κ value was never less than 0.85.

We used data from the video analysis to calculate the proportion of time allocated by the subjects to ‘states’ (behaviour duration/observation time) and the relative frequency of ‘events’ (number of events/observation time) for each 30 min focal observation. State behaviours were utilised to calculate the behavioural time budget.

We then analysed the data using IBM SPSS version 27 and RStudio v. 1.3.1093 [52]. We ran the Kolmogorov–Smirnov test to assess the distribution of the environmental and human-related parameters, and employed the Spearman’s *rho* correlation [53] to detect significant correlations between them (Rose et al. 2020). 

To test the effect of the selected parameters on behaviours, we ran generalised linear mixed models (GLMMs) on nine behavioural categories [18,19,20]). We selected the *resting*, *vigilance* and *out of sight* categories, as they were the most prevalent categories within the time budgets (see Section 3.1). We also selected the *ARBs*, *human–animal interaction*, *juggling*, *locomotion*, *swimming*, and *play* categories, corresponding to the main behavioural categories analysed in previous studies on captive ASCOs [30,35,36,37,38].

The proportion of time allocated to behavioural categories per session per day was coded as the dependent variable. We ran GLMMs using the R-function *glm* (family = *beta*) of the R-package *glmmTMB* with logit links [54]. The beta distribution is commonly employed to model continuous proportion data [55]. Since proportion data are limited to numerical values between 0 and 1, we compressed the data range using the following equation: *p*◦ = (*p*(*n* − 1) + 1/2)/*n*, where *p* is the original proportion, and *n* is the sample size [56].

For the categories *resting*, *vigilance*, *out of sight*, *ARBs, human–animal interaction*, *juggling*, and *locomotion*, we coded individual ID (factorial: female, male), time of day (factorial: S1, S2, S3, S4, S5, S6), THI (numeric), background noise level (numeric), time spent by caregivers inside/around the enclosure (numeric), visitor score (numeric), and the interaction between background noise level and visitor score as fixed factors. We selected the interactions based on the results of the Spearman’s *rho* correlation [18]. We also included water temperature and the interaction between THI and water temperature as fixed factors for the categories *swimming* and *play*. Dates were coded as random factors in all the GLMMs. Weather condition and wind speed were excluded from the analysis because the weather was classified as ‘sunny’ on 12 out of 14 sampling days and wind was below the detection level in 83 out of 84 observation sessions.

We used a likelihood ratio test to detect if there was a statistically significant difference between the full and the null models (ANOVA with argument *Chisq*) [57]. In the case of significant factorial predictors, we ran the Tukey test (R-package; *multcomp*) to perform all pairwise comparisons, adjusting the level of probability with the Bonferroni correction [53,58]. When we found the significance of the interactions (noise ∗ visitor score, THI ∗ water temperature), we considered only the effect of the interaction. All statistical tests were two-tailed, with the significance level set at *p* < 0.05.

## 3. Results

### 3.1. Behavioural Time Budgets and Abnormal Repetitive Behaviours

As reported in Table 2, the most prevalent behavioural category in the otters’ time budget was *out of sight* (female: 40.48%; male: 42.70%). In particular the female spent 25.63% of the time in the indoor area (*out of sight_indoor_*), 12.005% in the outdoor section (*out of sight_outdoor_*), and 2.85% in the dens (*out of sight_dens_*). Similarly, *out of sight_indoor_* accounted for 31.43% of the male’s time budget, followed by *out of sight_outdoor_* (8.95%), and *out of sight_dens_* (2.32%). The next most common categories were *vigilance* and *resting*, corresponding, on average, to 16.70% (female = 15.88%; male = 17.51%) and 10.05% (female = 7.93%; male = 12.17%), respectively.

The category *feeding* accounted, on average, for 5.91% (female = 5.92%; male = 5.90%), of the behavioural time budgets, followed by *locomotion* (female = 3.52%; male = 3.86%), *swimming* (female = 3.42%; male = 3.86%), and *play* (female = 4.85%; male = 3.40%). We did not record any mating attempts from the male, and each of the remaining categories individually varied from a minimum of 0.03% (*other*) to a maximum of 2.62% (*exploration*) in the female’s time budget, and from a minimum of 0.01% (*agonistic*) to a maximum of 2.40% (*scent marking*) in the male’s budget.

Human–animal interactions were limited (female = 1.04%; male = 1.54%) and involved caregivers (female = 12.72 min, 0.54%; male = 14.19 min, 0.59%), visitors (female; 6.70 min, 0.28%; male = 16.69 min, 0.69%), and the observer (female; 5.30 min, 0.22%; male = 6.18 min, 0.26%). As for the behaviours directed towards people, the otters observed the caregivers, the observer and the visitors for 10.19 ± 0.34 (mean ± SE), 3.30 ± 0.84 (mean ± SE), and 5.56 ± 1.90 min (mean ± SE) and tried to approach them for 3.05 ± 0.19 (mean ± SE), 2.44 ± 1.28 (mean ± SE), and 6.14 ± 3.09 (mean ± SE) min, respectively, mainly swimming towards the pool’s glass viewing panel (10.63 ± 4.69 min, mean ± SE). We also recorded the male retreating from caregivers working inside the enclosure (0.41 min).

The otters could freely choose where to perform behaviours for most of the observation period (female = 96.83%; male = 96.68%). Access to specific parts of the enclosure was restricted only during the second daily feeding, to allow for husbandry procedures. Indeed, the caregivers locked the subjects in the outdoor area to clean the dens (female = *begging_outdoor_*, 29.91 min, 1.26%; male = *begging_outdoor_*, 20.86 min, 0.87%), then locked and fed the otters inside the dens to access the rest of the enclosure (female = *eating_dens_*, 45.14 min, 1.91%; male = *eating_dens_*, 58.94 min, 2.45%). Considering *ARBs*, we found the male begging for 21.54 min and the female for 36.72 min, amounting to 0.90% and 1.55% of their behavioural time budgets. Since these behaviours were performed only in the presence of the staff, they could also be counted within the category *human–animal interaction*, increasing the interaction with caregivers to 49.44 min (2.09%) for the female and to 35.73 min (1.49%) for the male. Both subjects also performed juggling, with the male juggling 14.89 min (0.62%) and the female 155.47 min (6.57%).

Concerning the behaviours classified as ‘events’ (Table 1), descriptive analysis showed that the female yawned 128 times (3.25 yawns per hour) and performed agonistic behaviours 19 times (0.48 occurrences per hour). Yawns were more frequent during the first observation session (S1 = 90; S2 = 13; S3 = 3; S4 = 3; S5 = 13; S6 = 6), whereas agonistic behaviours were most frequent during the first, fourth and sixth sessions (S1 = 4; S2 = 1; S3 = 1; S4 = 5; S5 = 2; S6 = 6). We counted 112 yawns (2.80 per hour) emitted by the male, and 20 occurrences of agonistic behaviours (0.50 per hour). We mainly recorded yawns in the first and second observation sessions (S1 = 43; S2 = 38; S3 = 3; S4 = 5; S5 = 16; S6 = 7). Agonistic behaviours mostly occurred in the fifth session, and were absent in the third one (S1 = 3; S2 = 2; S3 = 0; S4 = 2; S5 = 10; S6 = 3).

### 3.2. Environmental Conditions and Human Presence

Across the 14 sampling days, the ambient temperature fluctuated between 19.00 and 34.30 °C, with a mean (±SE) of 28.03 ± 0.25 °C. Humidity values ranged from 40 to 85%, resulting in a mean (±SE) of 60.75 ± 0.93%. The THI values fell within the range 19.00–34.30 (mean ± SE = 28.03 ± 0.25). The minimum and maximum values of water temperature were 18.20 and 28.80 °C, while the mean value (±SE) was 24.52 ± 0.17 °C. Weather and wind speed were excluded from the analysis, as reported in Section 2.4.

The median of the number of visitors, expressed as visitor score (Section 2.3.3), was 2 (i.e., 11–20 people). Background noise level ranged from 47.90 dB to 71.69 dB (mean ± SE = 62.17 ± 0.37 dB). Concurrently, caregiver presence lasted approximately six hours during the observation sessions. Significant positive correlations were found between THI and water temperature (Spearman’s rank: rs (168) = 0.585, *p <* 0.001) and between background noise level and visitor score (Spearman’s rank: rs (168) = 0.483, *p <* 0.001).

### 3.3. Integrated Effect of Selected Parameters on Behaviours

The full model for *resting* (GLMM_1_) did not statistically differ from the null model (likelihood ratio test: *χ*^2^ = 11.174, *df* = 14, *p* = 0.43). As for *vigilance* (GLMM_2_), the full model was significantly different from the null one (likelihood ratio test: *χ*^2^ = 50.450, *df* = 14, *p* < 0.001). Time of day (session) was a significant predictor of the target variable (estimate = −2.48 ± 0.39, z value = −6.33, *p* < 0.001; Appendix A), with the subjects allocating more time to checking the environment in the first observation session (Figure 2a). The results of the Tukey test are reported in Table 3. We also found a *p* value close to significance level for THI (estimate = 0.056 ± 0.0287, z value = 1.965, *p* = 0.05, Appendix A). The otters also engaged less in vigilance with increased THI values (Figure 2b).

We did not detect any difference between the full and the null model of the category *out of sight_dens_* (GLMM_3_; likelihood ratio test: *χ*^2^ = 6.173, *df* = 14, *p =* 0.861), whereas the full models *out of sight_indoor_* (GLMM_4_) and *out of sight_outdoor_* (GLMM_5_) were significantly different from the null ones (GLMM_4_, likelihood ratio test: *χ*^2^ = 33.920, *df* = 14, *p* < 0.001; GLMM_5_, likelihood ratio test: *χ*^2^ = 71.357, *df* = 14, *p* < 0.001). Both categories were affected by time of day (*out of sight_indoor_*: estimate = 2.14 ± 0.50, z value = 4.27, *p* < 0.001; *out of sight_outdoor_*: estimate = −1.70 ± 0.37, z value = −4.59, *p* < 0.001; Appendix A). The otters mainly occupied the indoor area during the third observation session, and stayed out of sight in the outdoor section mostly in the fourth session. The results of the Tukey test are reported in Table 4 and Table 5, and in Figure 3 and Figure 4a. In addition, animal identity was a statistically significant predictor of *out of sight_outdoor_* (estimate = 0.394 ± 0.126, z value = 3.116, *p* < 0.01: Appendix A), with the female spending more time not visible (Figure 4b).

The full model for *locomotion* (GLMM_6_) significantly varied from the null model (likelihood ratio test: *χ*^2^ = 48.170, *df* = 14, *p* < 0.001), with the time of day significantly influencing the level of the target variable (session: estimate = −1.32 ± 0.36, z value = −3.65, *p* < 0.001; Appendix A). In particular, the subjects moved less during the third observation session (Figure 5). The results of the Tukey test are reported in Table 6. As for *swimming* (GLMM_7_), we only found an almost significant tendency for water temperature, with the subjects swimming more with increased temperatures (likelihood ratio test: *χ*^2^ = 20.500, *df* = 14, *p* < 0.05; water temperature: estimate = 0.064 ± 0.033, z value = 1.89, *p* = 0.06: Appendix A). Results did not show any significant difference between the full and the null model for *play* (GLMM8, likelihood ratio test: *χ*^2^ = 9.723, *df* = 14, *p =* 0.56).

Regarding human–animal interactions, the full models for *caregiver–otter interaction* (GLMM9) and *visitor–otter interaction* (GLMM10) were found to be significantly different from the corresponding null models (GLMM9, likelihood ratio test: *χ*^2^ = 32.628, *df* = 14, *p* < 0.001; GLMM10, likelihood ratio test: *χ*^2^ = 30.456, *df* = 14, *p* < 0.001). The level of interactions with caregivers was influenced by the time of day (estimate = −0.791 ± 0.253, z value = −3.129, *p* < 0.01; Appendix A, Figure 6a) and by the presence of the staff in and around the enclosure (estimate = 1.47 ± 0.25, z value = 5.88, *p* < 0.001; Appendix A, Figure 6b). The results of the Tukey test can be found in Table 7. Animal identity was a significant predictor of *visitor–otter interaction* (estimate = −0.239 ± 0.106, z value = −2.25, *p* = 0.02: Appendix A), with the male interacting more than the female (Figure 7).

We detected a significant variation between the full and the null model for *juggling* (GLMM_11_, likelihood ratio test: *χ*^2^ = 21.531, *df* = 14, *p =* 0.028), with the female juggling more than the male (animal identity: estimate = 0.801 ± 0.149, z value = 5.37, *p* < 0.001; Appendix A; Figure 8).

The full model for *ARBs* differ from the null one (GLMM_12_, likelihood ratio test: *χ*^2^ = 21.175, *df* = 14, *p =* 0.031). The otters begged more when staff were present in/around the enclosure (estimate = 0.888 ± 0.295, z value = 3.02, *p* < 0.05; Appendix A; Figure 9).

## 4. Discussion

### 4.1. Behavioural Time Budget

Both otters performed a wide array of species-specific behavioural patterns [25], with a limited occurrence of aggressive and avoidance behaviours, which are usually associated with negative emotional states, such as fear and anxiety [40]. Moreover, the presence of potential ARBs, which usually indicate poor welfare conditions, was limited (for a detailed discussion see Section 4.2). The otters could also exercise control and choice over the environment, choosing where to perform behaviours almost 24 h a day. In fact, they were locked in the dens or in the outdoor area for approximately only 6 min per day, to allow for cleaning procedures. 

Behavioural variety, the prevalence of behaviours linked to positive affects, and the opportunity to make choices are considered positive welfare indicators [11,59]. Indeed, they reflect environments and husbandry practices able to meet individual needs and preferences [5,59]. The results obtained suggest that the subjects were experiencing primarily positive welfare states during the observation period [40].

Focusing in detail on the otters’ time budgets, the dominant category was *out of sight*, accounting, on average, for 41.59% of the subjects’ time budgets. This is in accordance with the results reported for breeding pairs kept with offspring in outdoor naturalistic enclosures [30,35,36]. Conversely, pairs and triplets of siblings held in indoor enclosures with limited natural vegetation and shelters were reported to spend less time out of sight [37,39]. Thus, the results of this study seem to confirm that different and enriched hiding options are key requirements in captive settings, in line with husbandry guidelines [24]. Nevertheless, a larger sample size is needed to exclude the influence of group composition.

When visible, the subjects mainly engaged in resting and vigilance behaviour. The percentage of time allocated to resting was similar to that reported for breeding pairs in family groups [30,35,36]. Vigilance performance was, instead, slightly higher in comparison with the available literature, wherein ASCOs spent approximately less than 15% of their time budget checking the surrounding environment [37,39]. The otters engaged less in land locomotion and affiliative interactions compared to the breeding pairs of two family groups studied by Cuculescu-Santana et al. [30] and Owen [35]. This discrepancy could be due to the absence of offspring in this study that could have stimulated affiliative interactions, and the overall increased activity level of the adults involved in providing parental care (i.e., carrying, following, and feeding cubs) [24].

Notably, the family group analysed by Gothard [36] allocated 20% of its time budget to foraging and feeding, with a 13% increase when enriched with crickets and mealworms scattered in the enclosure. All the other studies, including the present research, reached a maximum of half of that percentage. Since the species spends 40 to 60% of its waking time foraging in the wild [41], a greater use of food-based enrichments [3] is highly recommended.

Implementing more olfactory and manipulative enrichments, such as spices and toys [3], is also suggested to increase activity and encourage exploration, as described in previous studies on captive otters [60] and different carnivore species [61,62]. Enrichment could also help in decreasing the time allocated to vigilance, thus reducing alertness, which could be linked to negative affective states, such as anxiety [40]. It is worth noting that since resting and affiliative behaviours were mostly observed in the video-recorded den, they have probably been underestimated, due to the lack of video cameras in the other den and in the indoor zone. Further studies are, therefore, needed to better assess these behaviours. Finally, regarding behaviours classified as events (Table 1), agonistic interactions were rarely recorded, indicating a lack of competition between the two otters. Yawning mainly occurred in the morning, and the number of yawns emitted by the subjects was similar. Since, in many mammal species [63], yawning is an indicator of anxiety, future research on a larger sample size could be useful to shed light on the function of this behaviour in *A. cinereus*.

### 4.2. Integrated Effect of Environmental and Human-Related Factors on Behaviours

Average daily humidity and water temperature complied with the recommended range, whereas ambient temperature exceeded the maximum temperature recommended (30 °C) in husbandry guidelines by four degrees Celsius [24,64]. Water and ambient temperatures were positively correlated, while the latter was negatively correlated with humidity. Contrary to our expectations, ambient temperature and humidity did not affect the otters’ behaviours, meaning that the animals were able to cope with environmental conditions, although ambient temperature ranged from 19.00 to 34.30 °C. The subjects only showed a tendency to swim more with higher water temperatures, and engaged less in vigilance behaviour with increased THI values, in accordance with the study by Cuculescu-Santana and co-authors [37]. Findings suggest that the presence of lush vegetation and the 24 h access to the shelters, pool, and stream allowed the otters to thermoregulate without reducing their overall level of active behaviours.

As hypothesised, we did not detect an influence of visitor numbers, noise or of visitor interaction on behavioural responses, confirming the finding of recent studies that highlighted how visitor presence tends not to be the main impact factor when other parameters are taken into consideration [17,18,19]. This study, to our knowledge, is the first one to analyse the effect of background noise on ASCOs in captivity. Despite the positive correlation found between noise level and visitor numbers, and the presence of playground devices for children, the otters did not seem to be affected by these factors.

Our results mainly showed an effect of time of day (i.e., observation session) on the individuals’ behaviours, as previously reported for other species [17,18,19,21]. In particular, they moved less and spent the greatest amount of time in the indoor enclosure during the third observation session—which occurred during the hottest phase of the day (12:15 pm–1:15 pm). This finding could be explained by the fact that the temperature inside the indoor area was 1 °C less than in the outdoor section of the enclosure. Considering the amount of time spent by the subjects in the indoor zone, it is important that this area is designed to provide the animals not only with a shelter but also with an enriched environment similar to the outdoor section to encourage the expression of a wide range of species-typical behaviours. In addition, the time the otters were out of sight in the outdoor zone varied across observation sessions with a peak in the fourth one (2:45 pm–3:45 pm).

Our results confirm the need to provide ASCOs with multiple options in terms of shelters and shaded areas, to allow the animals to select the best spots for thermoregulation according to their needs, which can vary throughout the day. Otters could also be provided with temperature-controlled areas, to ensure that they have at their disposal zones with temperatures falling in the range recommended by husbandry guidelines [24]. We also found that the otters were vigilant mainly during the first observation session (9:15 am–10:15 am), probably reflecting the timing of routine husbandry procedures in the nearby enclosures.

The otters involved in this study spent 20% less time begging compared to published research [36,37]. ASCOs have commonly been observed begging in the presence of animal care personnel and the visiting public [30,35,37]. Instead, the study subjects begged only when caregivers brought them food, mostly when they were locked in the outdoor area waiting to enter the dens to be fed. The performance of this behaviour was, thus, triggered and reinforced by the presence and subsequent delivery of food [35,36]. 

According to Gothard [36], begging is stimulated by hunger and frustration caused by the limited opportunity to express the appetitive components of feeding. To be classified as an ARB, begging should include the repetitive up and down movement of the forepaws and, through time and repetition, its occurrence should be detached from the eliciting stimulus [15]. When begging, the otters in this study did not show any repetitive movements of their paws, but only observed and followed their caregivers while vocalising. It seems, therefore, that the begging shown by the otters of this study cannot be yet defined as an ARB. Nevertheless, it must be carefully monitored, and husbandry must be reviewed to prevent the otters from waiting for food, and to avoid an escalation of the behaviour, which could lead to a state of anxiety and the onset of a condition of chronic stress [65].

Juggling (i.e., fast, erratic movements that pass an object between the forepaws and, sometimes, the mouth) could also be interpreted as an abnormal misdirected foraging behaviour, indicating welfare problems [38]. Both the otters of this study performed juggling, with the female dedicating significantly more time to this behaviour. Juggling occurred two hours after feeding, when otters are expected to be hungry [38], and immediately after. To classify juggling as a feeding anticipatory behaviour, the otters should allocate more time to the behaviour before than after feeding events. We could also expect an increase in juggling in case of a delay in food provision. However, since we did not find any difference in relation to the provision of food, our results conflict with the literature [38], highlighting the need to conduct additional studies to explore the function of this behaviour.

Preventive measures can be adopted to prevent and decrease food anticipation. For example, a balanced diet and foraging opportunities have been shown to diminish feeding anticipatory behaviours. Gothard [36] reduced begging by modifying the diet and stimulating foraging by throwing live insects in the enclosure. Hawke and colleagues [32] reduced locomotory pacing by using a catapult to scatter food items, while Ross [33] decreased self-directed hair plucking and door manipulation after providing food via grapevine balls (i.e., balls consisting of entangled grapevine with food hidden inside).

Individual identity was a significant predictor of the level of juggling, vigilance, and visitor–animal interaction, in accordance with results reported by other scholars studying bird species [18]. An analysis of the otters’ personality traits could be useful to better pinpoint the individual specific traits that should be considered when applying changes to the enclosure and to husbandry and care procedures. Indeed, personality influences how animals cope with their environment and interact with humans [66,67,68]. For example, bold individuals usually react to unknown stimuli by exploring them, whereas shy animals tend to become quiet and vigilant [68]. Furthermore, enlarging the sample size could be useful to investigate the potential influence of the sex and age of individuals. Available studies on time budgets have mainly detected variations in behavioural responses between adults and juveniles, with no remarkable differences between sexes [30,35,37]. Future studies involving more zoological institutions could help shed light on the effect of these factors on otter behaviour. Finally, we did not detect any effect of the selected factors on resting and play, with the otters allocating a similar amount of time to each other to inactivity and playful interactions.

### 4.3. Limitations

Due to the COVID-19 pandemic, the institution had to introduce a daily visitor cap, with a maximum of 700 people (i.e., 25% of the zoo’s capacity) visiting the park at the same time, to avoid any potential gatherings of people. Since these restrictions caused a 50% reduction in visitor attendance compared with the same period in 2019 [69], the impact of visitor presence on the animals’ behaviour could potentially change with more extreme visitor numbers.

During the observation period, the zoo did not have scheduled closing days. Therefore, it was not possible to include in the research design sampling days with no visitors, which could have served as a baseline against which to compare the visitor effect [70]. Furthermore, technical restraints did not allow additional video cameras within the indoor area and in the den D1 (Figure 3), reducing the visibility of the otters. Finally, the project focused on only two individuals kept at a single zoo; therefore, although results may be indicative, they cannot be reliably extrapolated to other otters in similar circumstances [70].

## 5. Conclusions

In conclusion, to our knowledge, this study was the first to investigate the integrated influence of environmental and human-related stimuli on the welfare of captive otters. Our results showed that the otters involved in the study mainly experienced positive welfare states, suggesting that they were provided with good welfare conditions. Enclosure design and husbandry practices were found to allow the subjects to perform a variety of species-specific behaviours, and to exert choice and control over their environment. Due to the small sample size, the results cannot be generalised, and the limited number of video recording devices might have led to an underestimation of resting and affiliative behaviours. The COVID-19 pandemic and associated restrictions caused a reduction in visitor numbers, meaning re-evaluation of the visitor effect in relation to other parameters taken into consideration in this study is warranted in the future. Moreover, further studies, aimed at investigating how the subjects’ behaviour changes during the day and across seasons, and varies with personality, are also recommended, to evaluate how otters’ needs change over time, and how this could alter visitor effects. This understanding could be used to optimise their husbandry routines and maximise positive experiences.

Despite the limitations acknowledged previously, this study increased the understanding of a scarcely researched topic, and confirmed the effectiveness of applying multivariate research methods to better analyse the array of lived experiences of captive animals. It also suggested suitable areas for further research and provided recommendations, such as the implementation of new enrichments, that could improve the welfare of the studied subjects.

## Figures and Tables

**Figure 1 animals-13-02228-f001:**
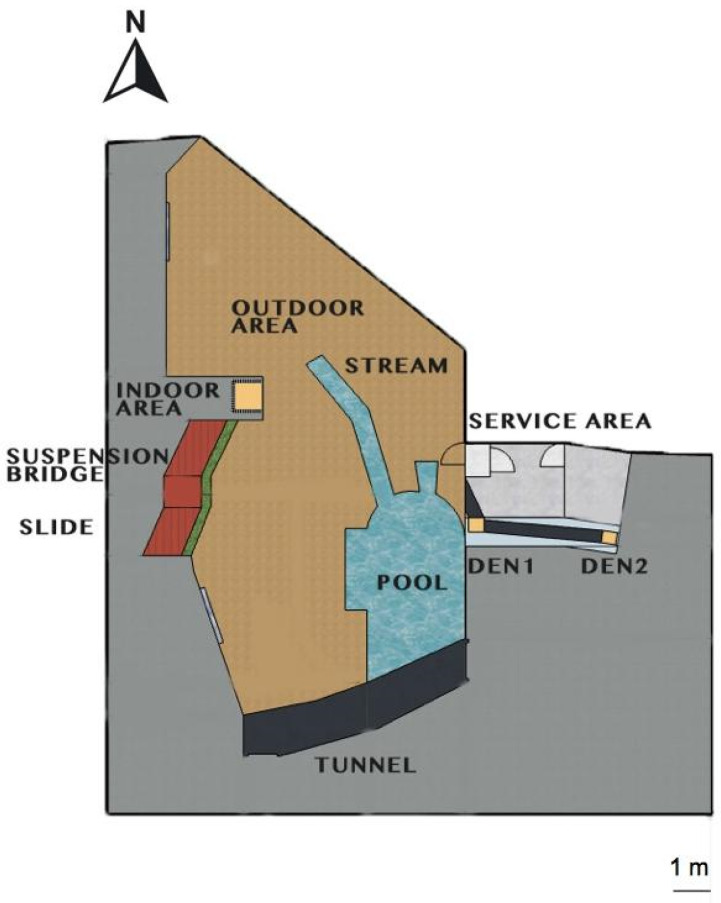
A schematic aerial diagram of the otters’ enclosure at GZP.

**Figure 2 animals-13-02228-f002:**
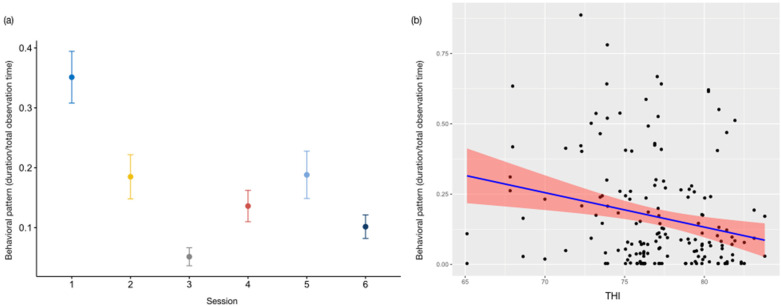
Variation in *vigilance* (*Y* axis) in relation to session ((**a**); *X* axis) and THI ((**b**); *X* axis). (**a**) The circle and bars represent the mean and 95% confidence intervals, respectively; Session 1: 9:15–10:15 am; session 2: 10:45–11:45 am; session 3: 12:15–1:15 pm; session 4: 2:45–3:45 pm; session 5: 4:15–5:15 pm; session 6: 5:45–6:45 pm; (**b**) the band represents the confidence interval.

**Figure 3 animals-13-02228-f003:**
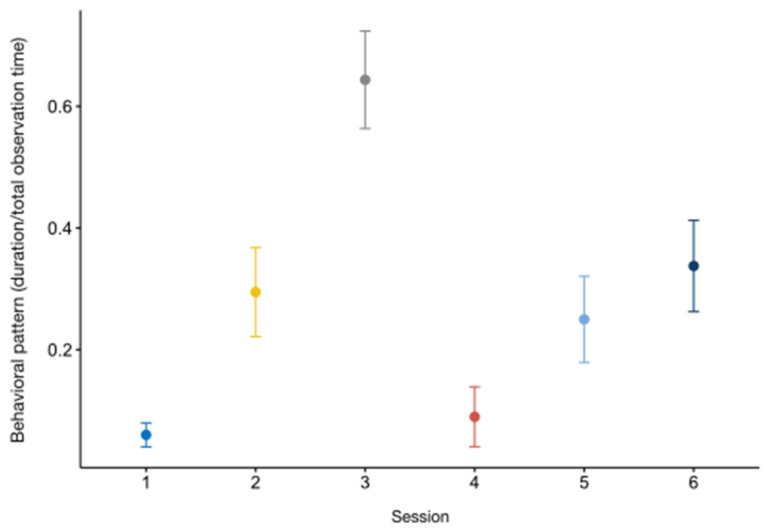
Variation in *out of sight_indoor_* (*Y* axis) in relation to session (*X* axis). The circle and bars represent the mean and 95% confidence intervals, respectively. Session 1: 9:15–10:15 am; session 2: 10:45–11:45 am; session 3: 12:15–1:15 pm; session 4: 2:45–3:45 pm; session 5: 4:15–5:15 pm; session 6: 5:45–6:45 pm.

**Figure 4 animals-13-02228-f004:**
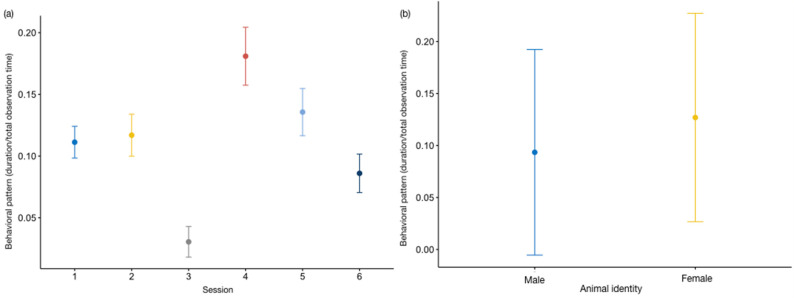
Variation in *out of sight_outdoor_* (*Y* axis) in relation to session ((**a**); *X* axis) and animal identity ((**b**); *X* axis). The circle and bars represent the mean and 95% confidence intervals, respectively. (**a**) Session 1: 9:15–10:15 am; session 2: 10:45–11:45 am; session 3: 12:15–1:15 pm; session 4: 2:45–3:45 pm; session 5: 4:15–5:15 pm; session 6: 5:45–6:45 pm.

**Figure 5 animals-13-02228-f005:**
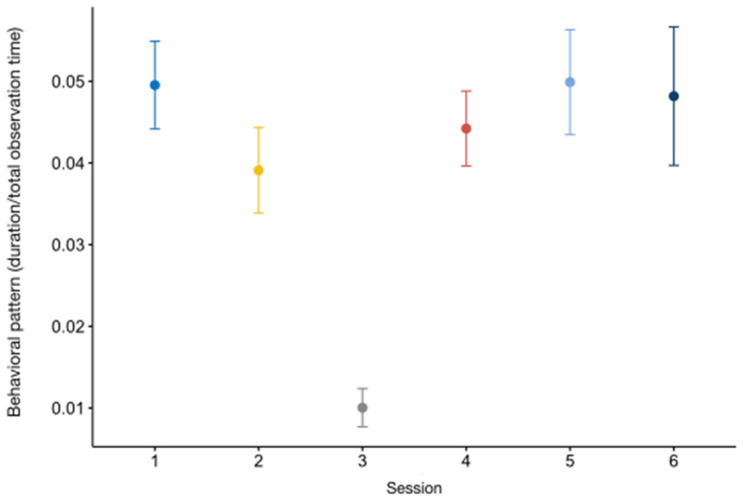
Variation in *locomotion* (*Y* axis) in relation to session (*X* axis). The circle and bars represent the mean and 95% confidence intervals, respectively. Session 1: 9:15–10:15 am; session 2: 10:45–11:45 am; session 3: 12:15–1:15 pm; session 4: 2:45–3:45 pm; session 5: 4:15–5:15 pm; session 6: 5:45–6:45 pm.

**Figure 6 animals-13-02228-f006:**
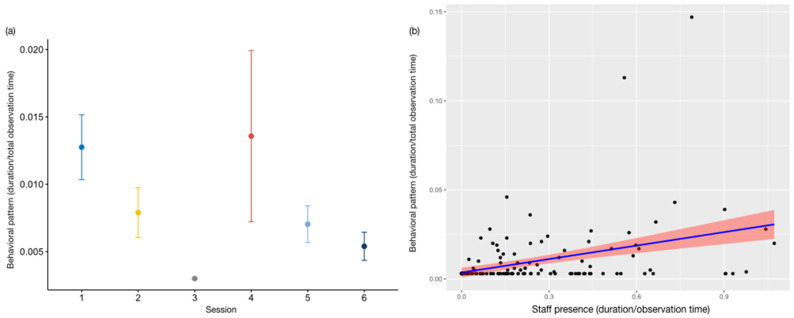
Variation in *caregiver–otter interaction* (*Y* axis) in relation to session ((**a**); *X* axis) and staff presence ((**b**); *X* axis). (**a**) The circle and bars represent the mean and 95% confidence intervals, respectively. Session 1: 9:15–10:15 am; session 2: 10:45–11:45 am; session 3: 12:15–1:15 pm; session 4: 2:45–3:45 pm; session 5: 4:15–5:15 pm; session 6: 5:45–6:45 pm. (**b**) The band represents the confidence interval.

**Figure 7 animals-13-02228-f007:**
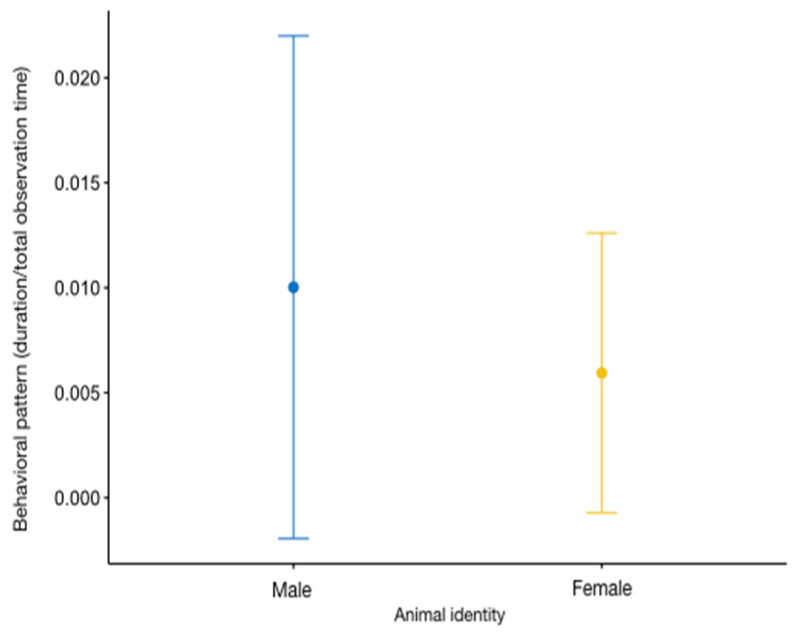
Variation in *visitor–otter interaction* (*Y* axis) in relation to animal identity (*X* axis). The circle and bars represent the mean and 95% confidence intervals, respectively.

**Figure 8 animals-13-02228-f008:**
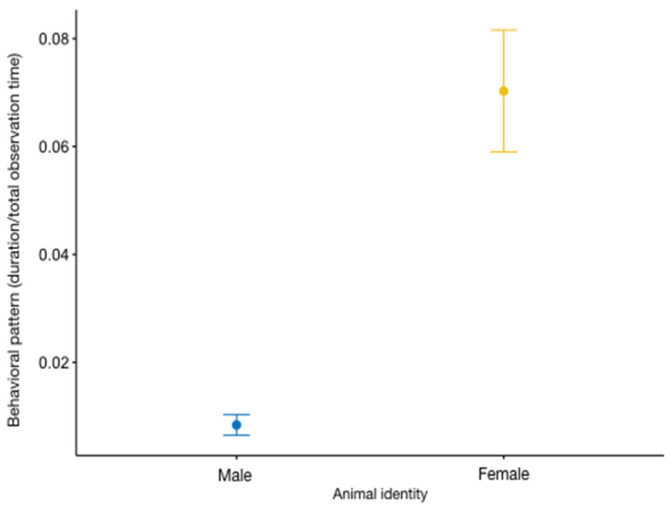
Variation in *juggling* (*Y* axis) in relation to animal identity (*X* axis). The circle and bars represent the mean and 95% confidence intervals, respectively.

**Figure 9 animals-13-02228-f009:**
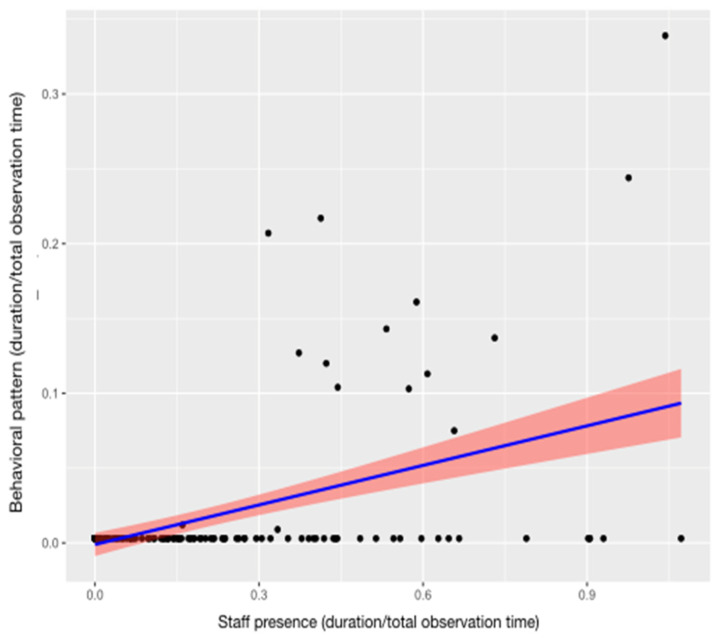
Variation in ARBs (*Y* axis) in relation to staff presence (*X* axis). The band represents the confidence interval.

**Table 2 animals-13-02228-t002:** The behavioural time budget of study subjects represented by the time (reported in minutes) and percentage dedicated to each behavioural category. The percentage was calculated as: (time allocated by the subject to a certain behavioural category or behaviour/the subject’s observation time during the study period) × 100.

Behavioural Category	Behaviour	Minutes and Percentage (%) by Behaviour				Minutes and Percentage (%) by Category			
		F		M		F		M	
ABRs	Begging Begging_outdoor_	6.82 29.91	(0.29) (1.26)	0.68 20.86	(0.03) (0.87)	36.72	(1.55)	21.54	(0.90)
Affiliative	Affiliative	15.30	(0.65)	34.04	(1.42)	15.30	(0.65)	34.04	(1.42)
Agonistic	Agonistic	0.00	(0.00)	0.16	(0.01)	0.00	(0.00)	0.16	(0.01)
Exploratory	Exploratory	61.93	(2.62)	20.56	(0.86)	61.93	((2.62)	20.56	(0.86)
Feeding	Eating/Foraging Eating_dens_	94.83 45.14	(4.01) (1.91)	82.90 58.94	(3.45) (2.45)	139.97	(5.92)	141.84	(5.90)
HAI	Caregiver Observer Visitor	12.72 5.30 6.70	(0.54) (0.22) (0.28)	14.19 6.18 16.69	(0.59) (0.26) (0.69)	24.72	(1.04)	37.06	(1.54)
Juggling	Juggling	155.47	(6.57)	14.89	(0.62)	155.47	(6.57)	14.89	(0.62)
Locomotion	Land locomotion	83.36	(3.52)	92.81	(3.86)	83.36	(3.52)	92.81	(3.86)
Maintenance	Maintenance	8.00	(0.34)	5.97	(0.25)	8.00	(0.34)	5.97	(0.25)
Mating	Mating	15.33	(0.65)	0.00	(0.00)	15.33	(0.65)	0.00	(0.00)
Nest building	Nest building	15.42	(0.65)	16.04	(0.67)	15.42	(0.65)	16.04	(0.67)
Other	Other	0.82	(0.03)	0.82	(0.03)	0.82	(0.03)	0.82	(0.03)
Out of sight	Out of sightin_door_ Out of sight_dens_ Out of sight_outdoor_	606.26 67.40 283.88	(25.63) (2.85) (12.00)	754.87 55.75 214.91	(31.43) (2.32) (8.95)	957.54	(40.48)	1025.52	(42.70)
Play	Locomotory play Object play Social play	18.31 5.82 80.87	(0.77) (0.25) (3.83)	0.00 0.38 81.06	(0.00) (0.02) (3.38)	114.69	(4.85)	81.44	(3.40)
Resting	Resting	187.50	(7.93)	292.27	(12.17)	187.50	(7.93)	292.27	(12.17)
Scent marking	Scent marking	57.01	(2.41)	57.65	(2.40)	57.01	(2.41)	57.65	(2.40)
Self-directed	Self-grooming Self-scratching	17.58 17.49	(0.74) (0.74)	24.14 21.43	(1.01) (0.89)	35.08	(1.48)	45.57	(1.90)
Swimming	Swimming	80.87	(3.42)	92.77	(3.86)	80.87	(3.42)	92.77	(3.86)
Vigilance	Vigilance	375.59	(15.88)	420.59	(17.51)	375.59	(15.88)	420.59	(17.51)

Note: ARBs = Abnormal Repetitive Behaviours; HAI = Human-Animal Interaction.

**Table 3 animals-13-02228-t003:** Full results of the Tukey tests for *vigilance*.

Predictors	Estimates	SEM	*χ* ^2^	*p*
*vigilance*				
S2 vs. S1	−1.698	0.349	−4.867	**<0.001**
S3 vs. S1	−2.479	0.391	−6.333	**<0.001**
S4 vs. S1	−1.719	0.375	−4.578	**<0.001**
S5 vs. S1	−1.603	0.356	−4.497	**<0.001**
S6 vs. S1	−1.683	0.312	−5.393	**<0.001**
S3 vs. S2	−0.781	0.319	−2.446	0.136
S4 vs. S2	−0−021	0.296	−0.070	1.000
S5 vs. S2	0.095	0.295	0.323	0.999
S6 vs. S2	0.015	0.307	0.050	1.000
S4 vs. S3	0.760	0.268	2.832	0.051
S5 vs. S3	0.876	0.268	3.272	**0.013**
S6 vs. S3	0.796	0.298	2.670	0.079
S5 vs. S4	0.116	0.259	0.447	0.998
S6 vs. S4	0.036	0.290	0.124	1.000
S6 vs. S5	−0.080	0.275	−0.290	1.000

Note: *p* < 0.05 is boldface.

**Table 4 animals-13-02228-t004:** Full results of the Tukey tests for *out of sight_indoor_*.

Predictors	Estimates	SEM	*χ* ^2^	*p*
*out of sight_indoor_*				
S2 vs. S1	1.034	0.462	2.237	0.215
S3 vs. S1	2.139	0.501	4.273	**<0.001**
S4 vs. S1	0.453	0.484	0.935	0.935
S5 vs. S1	0.846	0.464	1.824	0.442
S6 vs. S1	1.018	0.408	2.491	0.123
S3 vs. S2	1.106	0.419	2.642	0.085
S4 vs. S2	−0.580	0.390	−1.497	0.665
S5 vs. S2	−0.189	0.377	−0.496	0.996
S6 vs. S2	0.015	0.409	−0.035	1.000
S4 vs. S3	−1.687	0.357	−4.721	**<0.001**
S5 vs. S3	−1.293	0.360	−3.595	**0.004**
S6 vs. S3	−1.120	0.383	−2.928	**0.039**
S5 vs. S4	0.393	0.335	1.172	0.846
S6 vs. S4	0.566	0.371	1.525	0.641
S6 vs. S5	0.172	0.360	0.479	0.997

Note: *p* < 0.05 is boldface.

**Table 5 animals-13-02228-t005:** Full results of the Tukey tests for out of *sight_outdoor_*.

Predictors	Estimates	SEM	*χ* ^2^	*p*
*out of sight_outdoor_*				
S2 vs. S1	−0.588	0.313	−1.881	0.400
S3 vs. S1	−1.699	0.370	−4.590	**<0.001**
S4 vs. S1	−0.003	0.347	−0.090	1.000
S5 vs. S1	−0.460	0.329	−1.397	0.717
S6 vs. S1	−0.819	0.280	−2.922	**0.038**
S3 vs. S2	−1.110	0.288	−3.856	**0.001**
S4 vs. S2	0.556	0.244	2.283	0.192
S5 vs. S2	0.128	0.241	0.530	0.994
S6 vs. S2	−0.231	0.268	−0.861	0.952
S4 vs. S3	1.667	0.243	6.848	**<0.001**
S5 vs. S3	1.238	0.245	5.048	**<0.001**
S6 vs. S3	0.879	0.277	3.171	**0.018**
S5 vs. S4	−0.429	0.207	−2.071	0.291
S6 vs. S4	−0.788	0.254	−3.106	**0.022**
S6 vs. S5	−0.359	0.245	−1.465	0.674

Note: *p* < 0.05 is boldface.

**Table 6 animals-13-02228-t006:** Full results of the Tukey tests for *locomotion*.

Predictors	Estimates	SEM	*χ* ^2^	*p*
*locomotion*				
S2 vs. S1	−0.567	0.288	−1.964	0.341
S3 vs. S1	−1.323	0.363	**−3.650**	**0.003**
S4 vs. S1	−0.278	0.404	−0.687	0.981
S5 vs. S1	−0.306	0.316	−0.967	0.919
S6 vs. S1	−0.475	0.290	−1.635	0.549
S3 vs. S2	−0.757	0.266	−2.844	**0.046**
S4 vs. S2	0.289	0.276	1.050	0.888
S5 vs. S2	0.261	0.220	1.185	0.827
S6 vs. S2	0.093	0.236	0.392	0.999
S4 vs. S3	1.047	0.236	4.430	**<0.001**
S5 vs. S3	1.018	0.218	4.666	**<0.001**
S6 vs. S3	0.850	0.253	3.491	**0.006**
S5 vs. S4	−0.029	0.220	−0.130	1.000
S6 vs. S4	−0.197	0.243	−0.810	0.960
S6 vs. S5	−0.168	0.206	−0.816	0.959

Note: *p* < 0.05 is boldface.

**Table 7 animals-13-02228-t007:** Full results of the Tukey tests for *caregiver–otter interaction*.

Predictors	Estimates	SEM	*χ* ^2^	*p*
*caregiver–* *otter interaction*				
S2 vs. S1	−0.791	0.253	−3.129	**0.021**
S3 vs. S1	−0.432	0.278	−1.555	0.620
S4 vs. S1	−0.136	0.280	−0.486	0.997
S5 vs. S1	−0.381	0.250	−1.521	0.642
S6 vs. S1	−0.460	0.216	−2.134	0.262
S3 vs. S2	0.359	0.259	1.385	0.728
S4 vs. S2	0.654	0.251	2.601	0.094
S5 vs. S2	0.410	0.219	1.867	0.414
S6 vs. S2	0.330	0.244	1.354	0.748
S4 vs. S3	0.296	0.203	1.453	0.686
S5 vs. S3	0.051	0.217	0.247	1.000
S6 vs. S3	−0.029	0.245	−0.116	1.000
S5 vs. S4	−0.244	0.207	−1.183	0.840
S6 vs. S4	−0.324	0.243	−1.336	0.758
S6 vs. S5	−0.080	0.220	−0.364	0.999

Note: *p* < 0.05 is boldface.

## Data Availability

The data presented in this study are available on request from the corresponding author.

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
