# Peer review of "The Integrated Effect of Environmental Conditions and Human Presence on the Behaviour of a Pair of Zoo-Housed Asian Small-Clawed Otters"

_animals, 2023, doi:10.3390/ani13132228_

Round 1
Reviewer 1 Report
Dear authors,
Thank you for the opportunity to review this work. I thought it was interesting, and I certainly agree that factors affecting zoo animals are multifaceted and we need to work to understand better ways of acknowledging their individual and combined impacts.
I have only minor comments on this work, but I think the manuscript will be improved if there is clarification in some areas:
L153 – 155: can you expand with thoughts on why this might be the case – it would then feed nicely into you looking at temp/humidity within the work
L210: if these things are there all the time they technically cease to become enrichment, they are enclosure design. I would change this to say ‘including’ or something like that.
Ethogram: you categorised some behaviours according to their location – I’d like to see a rationale for this as to why they were separated out, particularly when it came to the analysis. Does OOS include when they were locked in specific areas, or were observations only undertaken when they had free access?
Section 2.3.2: please clarify how much of each area could be seen on the cameras and also what access they had and when
L251: am I then right in thinking that you watched one animal for 30 mins and then the next, if this is the case then please clarify how you chose the order
L254: why did these obs periods differ?
L264/265: how were the environmental variables measured? Was it done in person? If so, could this have then affected the otter behaviour?
Section 2.4: is observation time the full 30 mins minus time there was no recording? Please clarify this. Did you combine or leave separate your behavioural subcategories? Please also clarify this, and the rationale for leaving separate if you did (as per above).
L314 – 316: briefly expand as to why these were excluded
Section 3.1: it seems odd to include ‘no recording’ within the behavioural results. I would have this just as an overarching sentence at the beginning instead. So the reader knows straight away what the max possible time spent in a particular behaviour could be.
L341: what other types of HAI could there have been? This reads like there are more.
L348 – 351: this needs including in the methods
L355 – 357: why did you record these behaviours separately, if you then think they should be put together as they may be one and the same?
L359: juggling has reasonably high %, why is this not incl in para 2?
L362 – 368: is this descriptive or inferential? If descriptive then state that, if inferential then it needs stats outputs
L371: typo on subject
L385: this would make more sense as a median value as you used an ordinal scale
Results table 3/4 would be better either coming after all the relevant text or (and on reflection I think this may work better) being broken down into separate tables per behaviour. I found myself doing lots of flicking back/forth when I got to different paragraphs.
Figure 2: x axis is individual I think?
For all the figures – make it clear what the time period is on the y axis – per session, per day, total time? throughout the MS you use % of whole obs so that might make sense for consistency
L474: you didn’t seem to technically look at ‘avoidance’, unless you consider moving away from the other animal to be avoidance. It’s a part of avoidance, but they may also just be keeping their distance
L487 - 496: somewhere it would be useful to clarify how much of your OOS is due to the camera positioning, rather than animals being out of public view? I agree that them being out of public view engaging in positive behaviours probably = better welfare, as the animals have a chance to be off show. But if they are scared/hiding or its just the cameras not picking them up, then this is a bit different so you would need to tone down that aspect of the discussion.
L509 – 518: I don’t understand why enrichment is being pushed here, you didn’t specifically look at this. I think the main aim of your study (i.e. to understand the multifacted nature of animal responses to their environment) is more important and needs focusing on
L552 – 554: it might also be worthwhile to consider providing appropriate temperature zones for animals – i.e. that the water is the correct temp, or that they have cooler (temp controlled) areas etc
L567: if begging is occurring before feeding this is anticipatory, not a stereotypy (your last few words of that sentence support this)
L622: I don’t think you underestimated this per se, as you categorised visitor number, but your results could potentially change if there were more extreme visitor numbers
L642: this work needs undertaking in relation to the other parameters you highlighted
L644: how otters needs change over time, and how this interacts with visitors
Reviewer 3 Report
The paper explores the effect of environmental parameters including impacts of people on the behaviour of a captive pair of Asian short clawed otter. While there is some merit in the methods used to analyse the data, and as a case study of these individuals the work provides further research on captive ASCOs, the work is rather limited in application to the wider field of captive research. With only 42 hours of data collection per individual and almost half of this time the animals were out of side with no opportunity to observe their behaviour during this time, I feel the interpretations are limited. The paper attempts to analyse the effects of visitor interactions, including caretaker interactions but this was observed for only a very small proportion of the time. While the paper is written well, I don't feel the work is focused enough on welfare to justify the title as this is entirely focused on behaviour which is only one aspect of welfare.
Some additional points of query are below:
No evidence of intra- reliability tests from the video footage.
Rather than being a stand alone category, out of sight should be accounted for in the overall proportions of the behaviours observed.
I don't understand why- no recording, was recorded. Why not start the sessions after this readjustment, given that it took up 10% of all the observation time.
Round 2
Reviewer 1 Report
Dear authors,
Thank you for taking the time to respond to my comments and make the revisions. I think there is still a slight lack of clarity in the reporting of the results (possibly as a result of you trying to juggle requests from lots of reviewers), but I think if those can be ironed out it will be a great piece of work.
Results comments in general:
Whenever you include significant statistical values (e.g. L401 – ToD was a significant predictor…) you need to include the full stats output, especially now that table has moved to the supplementary material
Table 3 (the post hoc tests) still has the same issue for me as the old table did. Being all together in one place makes it difficult to follow this through the rest of the text. I think it needs to be one table per analysis and then have each table after the paragraph in which it is referred
The male/female graphs would be clearer if labelled as male/female not 0/1
I also wonder if the time period graphs would be better that way too. If it would make it too cluttered then maybe put those in the figure legend, just to remind the reader. I spent a lot of time wondering what was happening at TP3, only discovered when I got to the discussion.
Line by line:
L30: as you have changed the title you probably should change welfare to behaviour
L85: where you have included all those new species examples you technically need species binomial names. But you refer to them straight after and it would also make it very cluttered. So I wonder if here do you need that species list, could you instead just have the references?
L254: last sentence is a repeat of earlier. But I am confused by this – alternating the subjects won’t help to monitor them across all time periods – because you were recording both of them across all time periods anyway
L317: I mentioned this previously but you need to say why they were excluded, not direct the reader to a place they haven’t yet got to. I think it would make sense to have that information here rather than later on.
Section 3.1.: it would be useful to say here how long the otters were locked inside for
L338/339: and the remaining… this doesn’t make sense, as you go on to cover behaviour which occurred for longer periods of time
Para starting L342: needs SEs with the averages
L342: this still reads like there are more than visitors/staff that animals are likely to have HAIs with. I see in the table that you included observer interactions, which are missing from here. I think it might be better to rephrase it to get rid of ‘mainly involved’. Maybe saying they were limited, for females more interactions with staff than visitors, for males more equal. Etc. Something that doesn’t make it sound like there are endless types of people to interact with. I think you also need to recognise that some was directed towards the observer – because it would seem from reading the table that the observer was influencing their behaviour.
L390: is VaR variance? I have never seen it written like this before. I would also recommend SE for consistency. Or instead just have (range 11 – 20).
Fig 4: graphs are the opposite way round to the legend
L489: why do you have a citation for how long the otters spent OOS?
Para from L557 onwards: if they are increasing their time inside at this point is there a need for a recommendation on good quality (i.e. comparable to outside areas) indoor areas?
Para from 589: I would also expect there to be a marked escalation if food was later than the schedule
Para from L606: I don’t see how this is relevant to your work. I also think giving a cue may well then just shift the point of anticipation to the cue, rather than the event. I think this paragraph is better off gone, and instead the focus remains on the things you looked at – i.e. that there are more than just visitors affecting zoo otter behaviour.
Reviewer 3 Report
Thank you for your response and for your edits to your manuscript. The work is clearer now and I think there is some merit in the overall findings as a case study.
